# Methods of Assessing Degradation of Supercapacitors by Using Various Measurement Techniques

**Stanislaw Galla** [1,*] **, Arkadiusz Szewczyk** [1] **, Janusz Smulko** [1] **and Patryk Przygocki** [2]

[1]  Departments of Metrology and Optoelectronics, Faculty of Electronics, Telecommunications and Informatics, Gdansk University of Technology, ul. G. Narutowicza 11/12, 80-288 Gdansk, Poland; szewczyk@eti.pg.edu.pl (A.S.); janusz.smulko@pg.edu.pl (J.S.)

[2]  Institute of Chemistry and Technical Electrochemistry, Faculty ff Chemical Technology, Poznan University of Technology, Berdychowo 4, 60-965 Poznan, Poland; patryk.przygocki@put.poznan.pl

[*]  Correspondence: galla@eti.pg.edu.pl; Tel.: +48-58-347-21-40



**Featured Application: It can potentially be used in capacitor tests.**

**Abstract:** This article presents the qualitative analyses of the construction of supercapacitor samples. The analyses are based on the suggested thermographic measurements as well as the technique of testing the inherent noise of the investigated element. The indicated assessment methods have been referred to the currently used parameters for the qualitative evaluation of supercapacitors. The approach described in this paper, which introduces additional parameters assessing worn out of supercapacitors, can be included in the so-called non-invasive measurement methods, which allow the assessment of the condition of the sample under test. This article presents the applied measurement stands and verifies of the applicability of measurement methods in relation to the currently used parameters allowing for the qualitative assessment of supercapacitors. The measurement method presented in this article was used to study prototypes of supercapacitors. The measurement results allow for more accurate characterization of the observed element. Conducted tests revealed, at the same time, that one of the proposed evaluation methods, based on measurements of inherent noise of tested supercapacitors, is a method predicting their degradation.

**Keywords:** supercapacitor; thermovision; noise; quality

## 1. Introduction

The massive application of supercapacitors in a variety of applications, as ancillary or even as major power sources, poses a variety of challenges. As a rule, constructors of such power systems try to use the maximum value of supercapacitors declared by the manufacturer, which has been described in a number of papers [1–4]. Theoretically, supercapacitors are characterized by a very long lifetime, however, they require a very special attitude as the declared values of parameters are to be restrictively approached. It is assumed that the basic parameters for assessing the degradation of supercapacitors are the changes in the capacitance $C$ and the equivalent series resistance $R_{ESR}$. As part of the project entitled "Mechanism of charging/discharging processes at the electrode/electrolyte interface of supercapacitor", additional methods were developed to support the qualitative assessment of worn out supercapacitors. These methods were based on noise and thermal measurements, which assume the use of existing and developed models of supercapacitors presented in [5–8]. Figure 1 presents the equivalent electrical circuit of the supercapacitor, using resistive and capacitive elements. It is the basis for other equivalent electric models of supercapacitors, which are presented in [7,8]. Most of the papers [9–13] were based on systems consisting of a series of n-consecutive RC branches. The

basic model (Figure 1) assumes the existence of two parallel branches of RC and the leakage resistance $R_L$, representing the leakage current of supercapacitor. The first branch $R_{ESR}$, $C_H$ consists of equivalent series resistance ($R_{ESR}$)—responsible for the main part of losses and the capacity $C_H$ of the Helmholtz layer, which are available for the rapid charging/discharging phase [14,15]. The second branch, $R_D$, $C_D$, represents the mechanism of charges redistribution occurring in supercapacitors, determining at the same time its velocity ($R_D$) and electric capacity of the diffusion mechanism ($C_D$) [10,15]. On the basis of the presented diagram, the criteria for qualitative assessment of supercapacitors are adopted. In classic assessment methods, they are represented by changes in electrical parameters: Capacitance $C$ measured between the terminals and $R_{ESR}$. It is assumed that the supercapacitor is completely degraded if the capacity $C$ decreased by at least 30% or the $R_{ESR}$ increased by 100% in relation to their initial values. Evaluation studies are carried out by the use of known methods described in [3,16–18]. Our paper presents the results of measurements and assessments of supercapacitor quality based on two measurement methods. The first method is focused on the measurement of temperature increments and the second utilizes measurements of noise generated in the tested samples. Reference is made to the experimental results of the proposed methods of the commonly used $C$ and $R_{ESR}$ values.

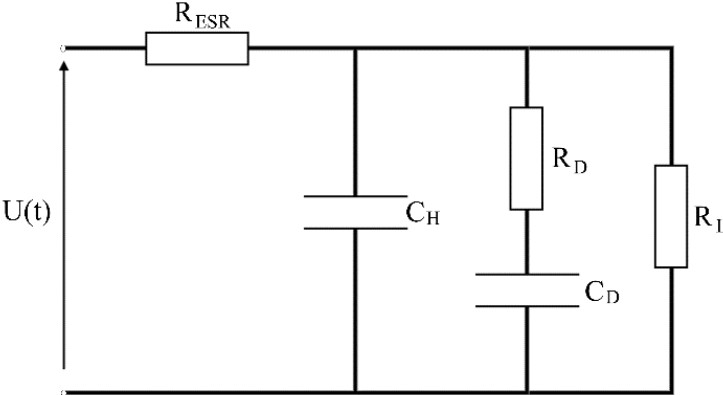

**Figure 1.** Equivalent electrical circuit of the supercapacitor.

The first proposed method is based on the analysis of temperature distributions on the observed surface of the supercapacitor during its intense charging/discharging. The basis of the evaluation presented in the paper are measurements of electromagnetic radiation in the infrared radiation (IR) range. The essentials and principles of thermographic measurements are described in more details, among others, in [19–22], which illustrate the conditions that should be taken into consideration while performing such type of measurements. The use of non-destructive measurement methods based on the use of the IR measurement technique is presented, inter alia, in [23,24]. Conducting thermographic tests of supercapacitors requires both proper preparation of the tested samples and consideration of a number of additional factors that may affect the measurement. The factors were presented in [25,26].

The second method utilizes noise measurements in the low frequency range. Noise is a well-known tool for determining the condition of electronic elements and materials. Noise level depends on working conditions as well as defects or degradation existing in the tested structures. It is often used to assess the quality of various objects and devices [27].

Noise in the range of low frequency can also be used as an indicator of the quality of capacitors [28] and supercapacitors [29,30]. The phenomena associated with the accumulation of charges in the supercapacitor, occurring at the at the electrode/electrolyte interface, are random processes. Changes in the intensity of these phenomena, caused by the degradation of electrodes or electrolytes, should affect the intensity of inherent noise generated within the investigated element.

## 2. Measurement Set-Up and Procedure

The suggested methods make use of the measurement set-up shown in Figure 2 (system for temperature distribution measurements) and in Figure 3 (noise measurement set-up). Both set-ups use the Atlas-Sollich power supply and measurement module (Multichannel Potentiostat Galvanostat) [31], controlled by the computer, collecting and processing all data. This module controls and registers current and voltage data parameters and further determines capacitance $C$ and $R_{ESR}$ values.

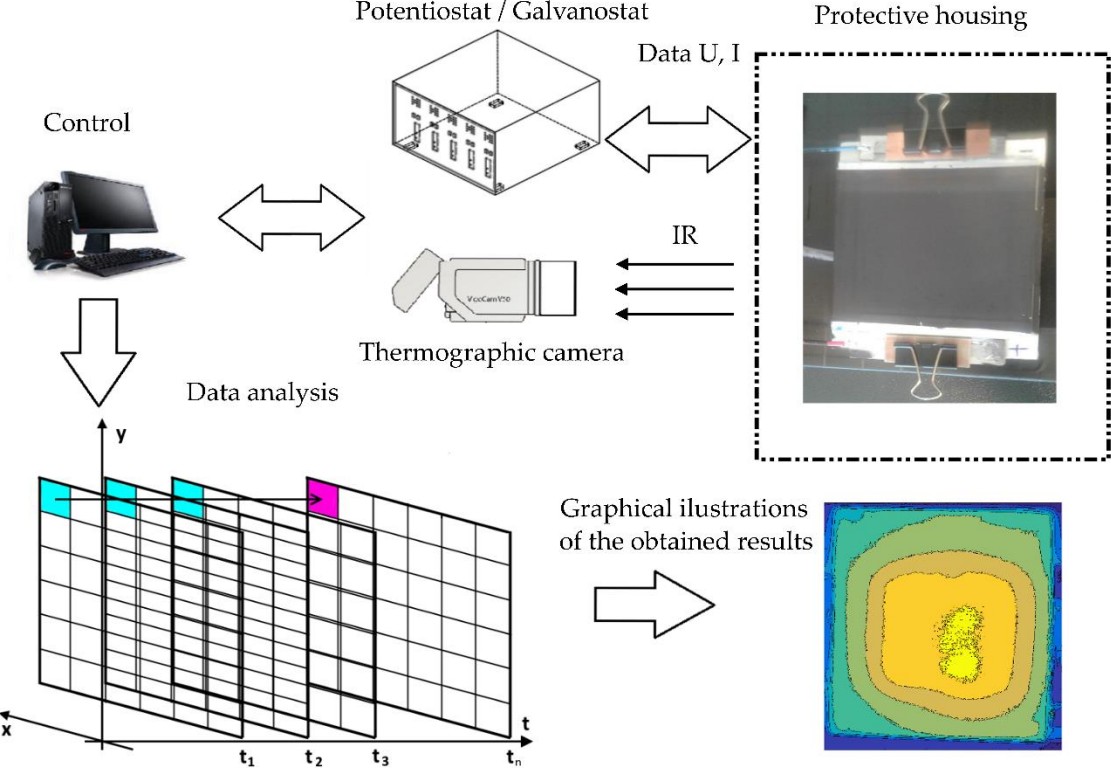

**Figure 2.** Graphical illustration of exemplary thermographic results.

The applied thermographic system for measurement of supercapacitors was presented in more detail in [26,27]. In the conducted research, no modifications were made to the measurement procedures proposed in [25]. The following are only the main measurement conditions for the applied thermovision system using the VIGOcam v5 thermal imaging camera (VIGO System S.A., Ozarow Mazowiecki, Poland). Its basic technical parameters are presented in Table 1.

**Table 1.** Basic parameters of VIGOcam v50 camera (adapted from [32]).

| Parameter | Value/Function Description |
|---|---|
| Detector type | Non-cooled bolometric matrix |
| Spectrum range | 8 ÷ 14 μm |
| Image resolution | 384 × 288 |
| Thermal resolution | ≤0.065 °C (for temperature 30 °C) |
| Field of vision | 15° × 11° |

In order to isolate test objects from external impacts, while taking tests, they were placed in a casing that was protecting them against direct effects of other external temperature fields. The inner surfaces of the protective casing were covered with a graphite layer in order to minimize reflections and to stabilize the emissivity of the background at a constant level. The tested samples were also coated with a graphite layer (Graphite 33 from Contact Cheme). In addition, a reference field was

introduced in the field of view of the camera. Thermographic measurements series of N thermograms were made in accordance with the time interval of 60 s as suggested in [27]. The obtained data was collected in a matrix consisting of N measurements having a size of 388 × 288 data of observed temperatures. The data on the thermographic assessment of tested samples was analyzed in an off-line manner according to the algorithm presented in [27]. It enabled the parameters supporting the assessment of the measurements to be obtained. The data presented in the paper refers to the determined values—maximum temperature increment ($\Delta T_{Max}$) of the observed surface during the charging/discharging process. The use of other parameters proposed in [27] for the evaluation of supercapacitors is presented in [26].

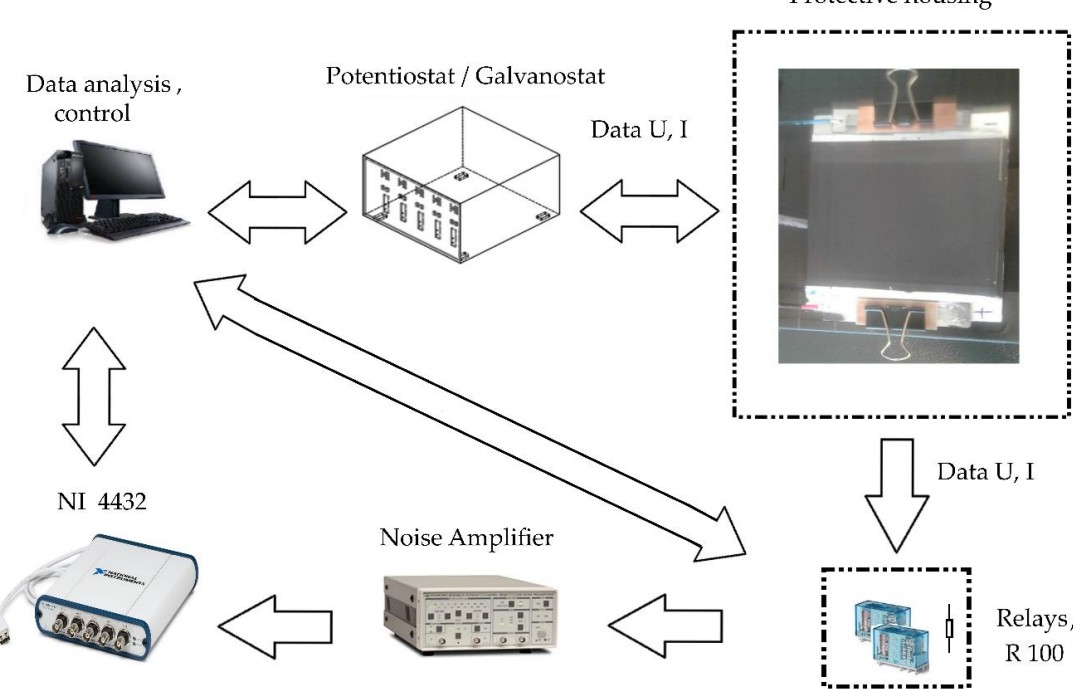

**Figure 3.** Block diagram of the measurement set-up applied for voltage noise time series recording.

The block diagram of the noise measurement set-up is shown in Figure 3. Analogously to the thermal imaging set-up, the Atlas-Sollich power supply module was applied to secure polarizing conditions. Low noise voltage preamplifier SR 560 was applied [33] and operated in the DC (DC coupling) range. The aim of the NI 4432 data acquisition board [34] was to sample noise time series using 24—bit analog-digital converter. A set of electrical relays disconnected the tested sample from the power supply unit when noise was recorded. This procedure eliminated interferences from the power supply unit. Additionally, the relays connected a loading resistor R = 100 Ω, applied to discharge the sample. The switching element is an electromagnetic relay ensuring stable connections, without eventual contact noise generation.

When the specimen was discharged, the additive noise component in the discharging current, determined by RC time constants of the tested specimen (Figure 1)—proportional to exponential decay as $e^{(-t/RC)}$ was observed. The voltage across the loading resistance which is proportional to the current supplied by the tested specimen was recorded. Loading resistance was selected to ensure a sufficiently long discharging time to record very low frequency components, even with fractions of Hz. The specimen was fully charged before our experiment. The charging time was long enough to secure charging of both capacities $C_H$ and $C_D$ (Figure 1). Otherwise, after disconnecting the power supply unit, the charges collected in the capacitance $C_H$ will flow to the second capacity $C_D$, which may induce false measurement results.

In our experiment, a 2 h period of charging using the constant voltage method preceded by charging with a constant current to a given voltage (1.5 V) was applied. The recorded discharging current is processed to remove a current of exponential decay and expose additive noise. Subsequently, the signal is divided into segments for which the power spectral densities are determined and then averaged to reduce random error. When the voltage across the supercapacitor's terminals reaches a value close to 0 V, the capacitor is reconnected to the power supply. Afterwards, the charging cycle for rated voltage ($U_c$) and discharge to ~ 0 V (approximately 10 mV) with constant charging/discharging current is repeated five times. The capacitance $C$ and the equivalent series resistance $R_{ESR}$ are determined based on the voltage waveform recorded during the fifth discharge. This repeated operation of charging/discharging stabilizes the specimen to ensure repeatable measurement results.

## 3. Methods of Electrical Parameters Evaluation

The presented methods of quality assessment of supercapacitors were subject to experimental studies. The validity of their use in relation to typical methods of assessing the degradation of the supercapacitor was evaluated. In the standard approach, the supercapacitor is assumed to be completely degraded (worn out) if the determined capacity decreases by at least 30% or $R_{ESR}$ increases by 100% in relation to the initial values. Figure 4; Figure 5 illustrate the methods used to determine $C$ and $R_{ESR}$. The methods of estimating $C$ and $R_{ESR}$ are the same for both measurements and concern the last five cycles, presented on the right side of the figures. The difference is visible at the first phase of treatment. Repeated charging/discharging is applied during temperature distribution measurements (Figure 4) and floating at the selected voltage $U_t$ =1.5 V is used before starting noise measurements (Figure 5).

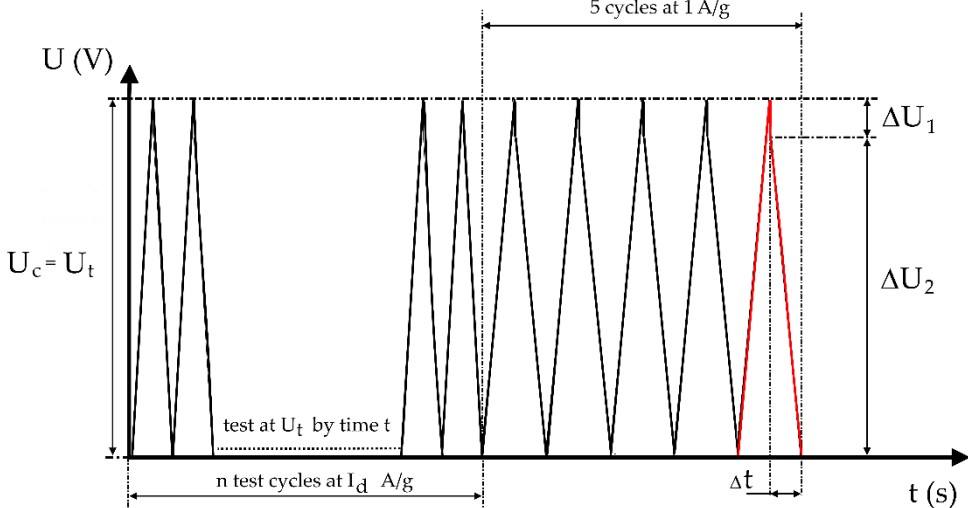

**Figure 4.** The applied way of determining values of capacitance $C$ and resistance $R_{ESR}$ after temperature distribution measurements.

The data obtained during the test allowed the estimation both $C$ and $R_{ESR}$ of the tested sample on the basis of the following Equations (1) and (2):

$$C = \frac{I_d \cdot \Delta t}{\Delta U_2} \tag{1}$$

$$R_{ESR} = \frac{\Delta U_1}{|I_d|}, \tag{2}$$

where:

$I_d$—test current (during charging/discharging process),

$\Delta t$—discharging time,

$\Delta U_1$—voltage drop across the resistance $R_{ESR}$, observed due to the change of current direction,

$\Delta U_2$—voltage drop during the discharge process at constant current.

The thermographic method was verified by investigating commercially available supercapacitors: Samxon, type DRL106SOTI25R (Man Yue Electronics Company Limited, Hong Kong, China). It can be indicated that degradation of the tested specimens by measuring the temperature increments $\Delta T_{MAX}$ as presented elsewhere [27]. Unfortunately, temperature increases after or during changes of the mentioned electrical parameters were observed. It can be concluded by considering the results presented in [26,27] that the temperature increase by 100% to the temperature increments observed for as-prepared samples indicates serious degradation of the tested supercapacitor. At the same time, the current density of the tested elements was estimated at the level of 4–6 A/g. In case of noise measurements, no verification tests were carried out on commercially available capacitors.

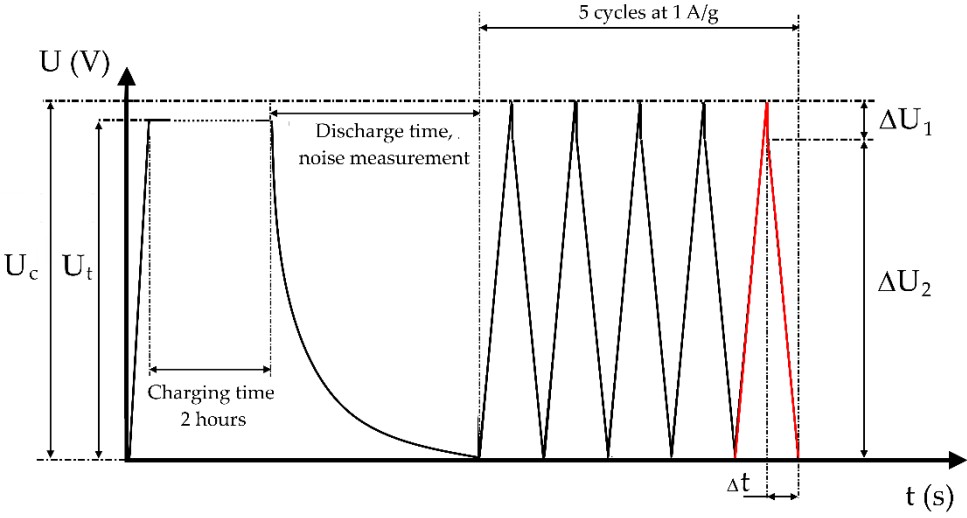

**Figure 5.** The applied way of determining values of capacitance $C$ and resistance $R_{ESR}$ after noise measurements.

## 4. Test Results of Prototype Samples of Supercapacitors

As part of the project, prototype samples of supercapacitors were subject to evaluation. They were made by technology based on aqueous electrolyte. The used samples of supercapacitors in measurements were presented in more detail in [35]. Subsequently, the manufactured supercapacitors were subject to forming and preliminary measurements in order to determine their electrical parameters: $C$ and $R_{ESR}$. As part of the aging test by carrying out charging/discharging processes, the measurements were performed at the adopted current density of 5 A/g (test current $I_d$ = 1225 mA) and the maximum 1.5 V charging voltage with $C$ and $R_{ESR}$ measurements performed every 250 cycles at current density 1 A/g (245 mA). A total of over 120,000 cycles were performed within the entire measurement period of the tested sample labelled P1. Measurements of $C$ and $R_{ESR}$ during the test are presented in Figure 6. Measurements applying the proposed noise method were also carried out for sample P1 when it was as-fabricated, after 48,000 cycles and after the completion of the research. For the tested sample P1, in the range up to 78,000 cycles, the maximum temperature increments of 2.5 °C was observed (at current densities of 5 A/g, test current $I_d$ = 1225 mA at test voltages from $U$ = 0.01 V to $U$ = 1.5 V). Exemplary distribution of temperature increments on the surface of the tested sample is presented in Figure 7. It illustrates the case of when the tested sample did not show changes of electrical parameters $C$ and $R_{ESR}$, indicative of deterioration processes. Changes in the measured values of electrical parameters occurred after 78,000 cycles. Figure 8 shows the distribution of the recorded

temperature increments after the completion of 80,000 cycles, when changes in the parameters $C$ and $R_{ESR}$ are visible.

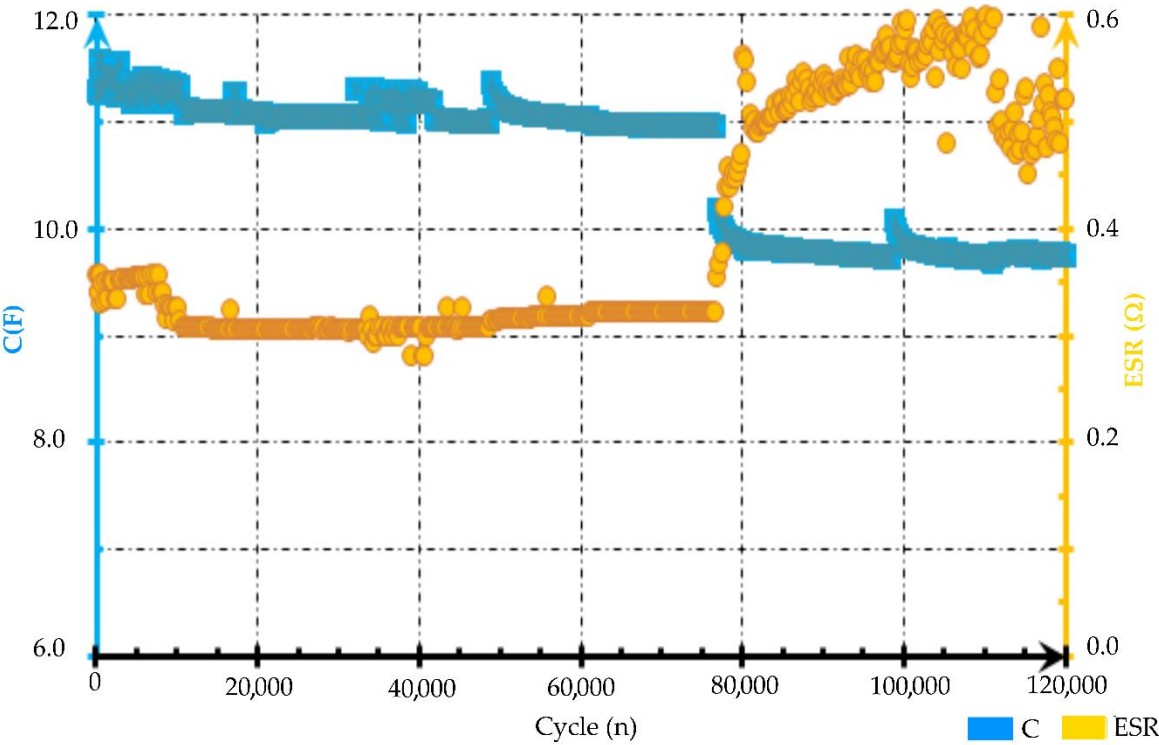

**Figure 6.** Capacitance $C$ and equivalent serial resistance $R_{ESR}$ recorded during the test of the investigated prototype sample at a current density of 5 A/g and polarizing voltage 1.5 V.

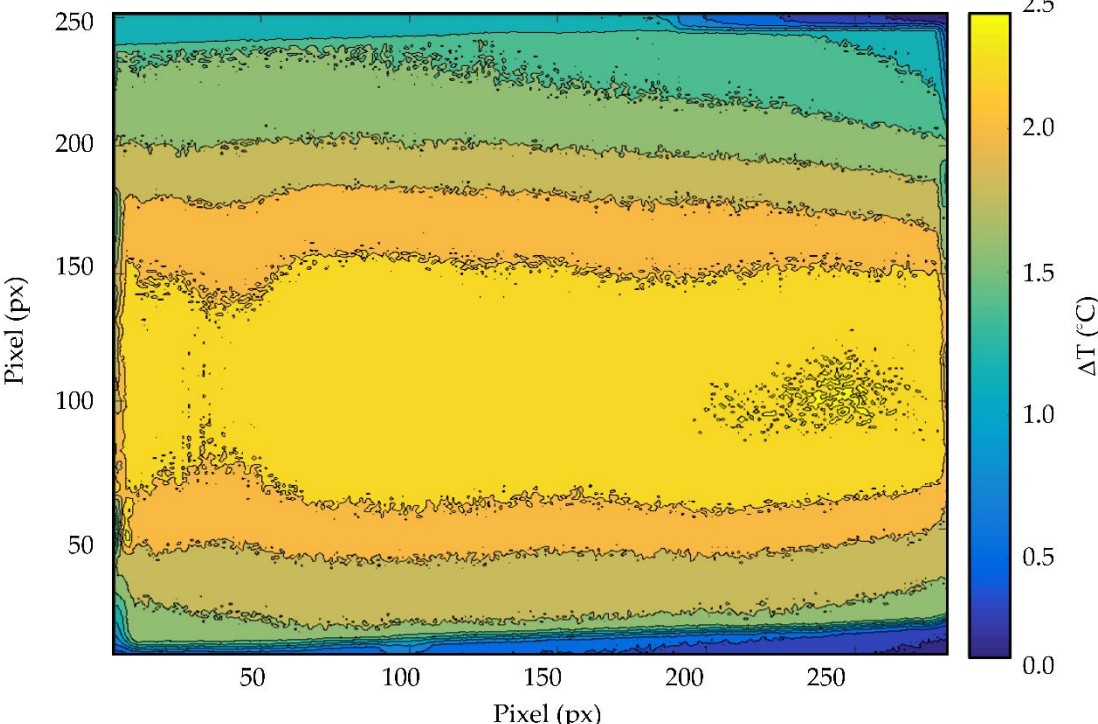

**Figure 7.** Illustration of distribution of temperature increases observed in the investigated prototype sample after 60,000 cycles of charging/discharging at a current density of 5 A/g and polarizing voltage 1.5 V; no serious changes of $C$ and $R_{ESR}$ were recorded.

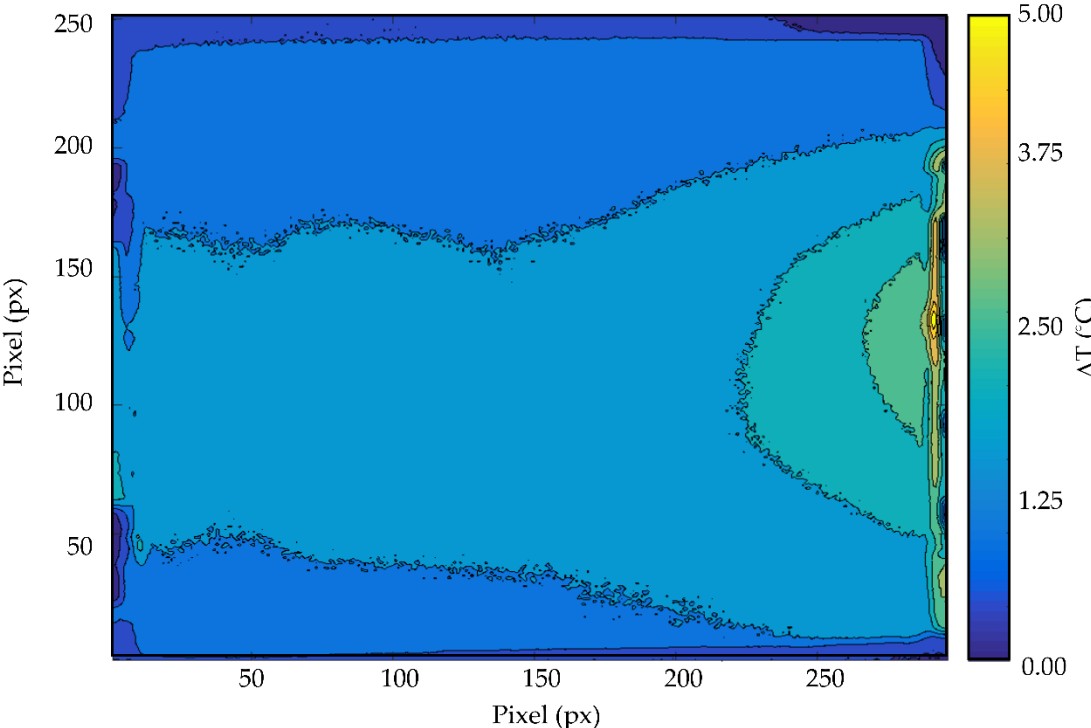

**Figure 8.** Illustration of distribution of temperature increases observed in the investigated prototype sample after 80,000 cycles of charging/discharging at a current density of 5 A/g and polarizing voltage 1.5 V; significant changes of $C$ and $R_{ESR}$ were recorded.

The minor increases in $C$ and $R_{ESR}$ capacities shown in Figure 6 after 48,000 cycles are the result of the interruption of aging tests (charging/discharging) and noise measurements. The change from continuous charging/discharging to floating at selected voltage induced some aging mechanisms and resulted in an accelerated drop of capacitance $C$ and an increase of resistance $R_{ESR}$. Similar performance was observed during the second time when noise was measured after approximately 98,000 cycles but of lower intensity. Preceding changes in the basic parameters $C$ and $R_{ESR}$ in the range up to 10,000 cycles most probably resulted from not fully completing the formation process of the supercapacitor.

Noise measurements of the tested prototype sample were performed by using the presented methodology. Noise was recorded for an as-prepared sample, followed by 48,000 and 120,000 charging/discharging cycles. The recorded values of capacitance $C$ and $R_{ESR}$ are presented in Table 2. Figure 9 presents the waveforms of the power spectral densities of voltage noise $S_{VC}(f)$ observed between its terminals during the discharging process. It has been concluded that the degradation of the tested sample is visible as an increase in its inherent noise level. Measurements indicate that change of noise intensity at white noise frequency range (between 2–20 Hz, Figure 8) occurs earlier than significant changes in the electrical parameters of supercapacitors and also earlier than changes in the distribution of the observed temperature increase.

**Table 2.** Electrical parameters capacitance $C$ and equivalent serial resistance $R_{ESR}$ of the tested prototype specimen.

| Time | C (F) | $R_{ESR}$ (mΩ) |
|---|---|---|
| As-fabricated | 11.2 | 350 |
| After 48,000 cycles | 11.1 | 320 |
| After 120,000 | 9.8 | 710 |

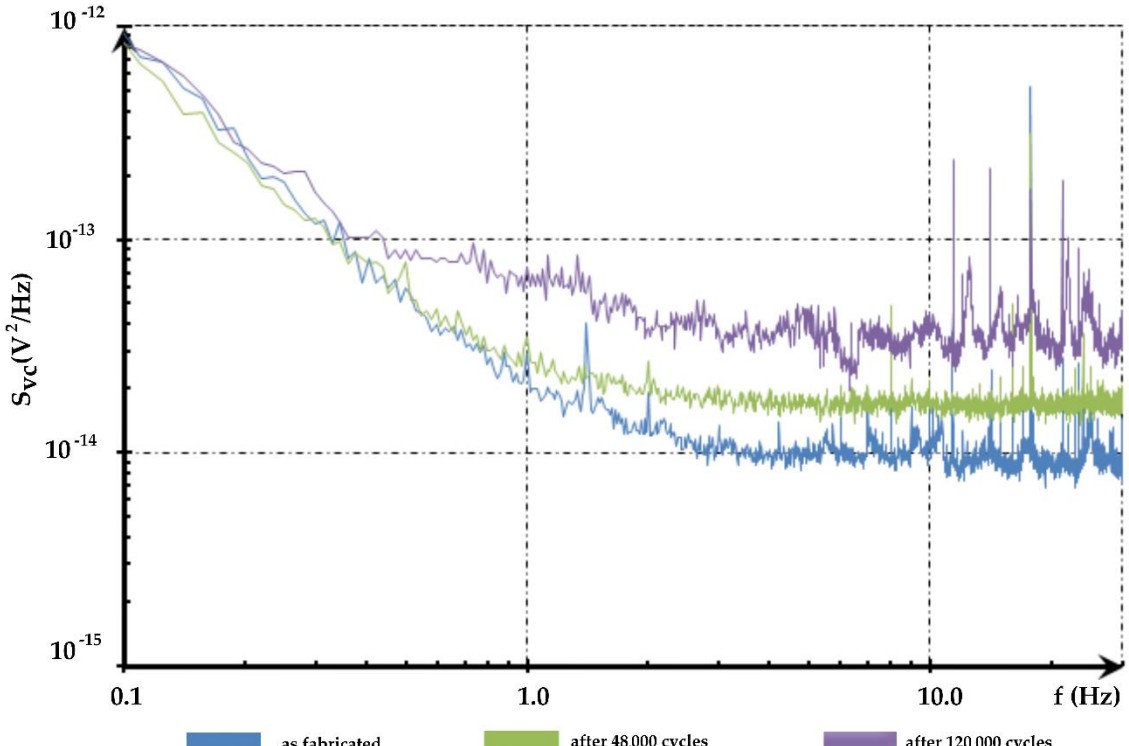

**Figure 9.** Power spectral densities of the identified voltage fluctuations across the tested prototype sample after selected number of charging/discharging cycles at a current density of 5 A/g and polarizing voltage 1.5 V.

## 5. Conclusions

The presented results indicate the possibility of applying other methods related to the distribution of temperature increases and noise intensity to evaluate the degradation of the supercapacitor. These methods can give supplementary criteria for qualitative evaluation, other than, or even instead of, industrial electrical parameters $C$ and $R_{ESR}$. There is a connection between the temperature increase and the changes occurring in the electrical parameters $C$ and $R_{ESR}$ of the supercapacitors. Temperature increases indicates degradation changes inside the supercapacitor. In the observed samples, changes of the resistance $R_{ESR}$ were accompanied by an increase in the determined temperature increment $\Delta T$ by over 100%. The observed changes of temperature are clearly correlated with the increase of $R_{ESR}$ but the distribution of temperature increases localized the areas where degradation was the most severe (local resistance represented by $R_{ESR}$ increased the most). This is important information to evaluate the quality of the applied construction, which may be useful in practice by monitoring temperature in critical points. This may be accomplished by applying relatively cheap temperature sensors.

As the presented, measurements show the increase of noise level indicates the occurrence of degradation processes. Noise increased earlier than the registered changes in capacitance $C$ and resistance $R_{ESR}$, as well as recorded temperature increases. This indicates using noise measurements to predict degradation processes in supercapacitors. More intense changes of noise levels were observed at the frequencies range 2–20 Hz than at lower frequencies where 1/f—like noise (flicker noise) prevails. The method has some drawbacks as it requires using specialized measuring set-ups and relatively long noise recording times. However, it can be definitely simplified in specific applications. Knowing the nature of noise changes for a wide frequency range, the measurement time can be shortened by limiting the measurements to selected band frequencies.

**Author Contributions:** Conceptualization, investigation, data acquisition, writing—review and editing S.G., A.S.; formal analysis J.S., samples making P.P.

**Funding:** This paper is financially supported by the National Science Centre Poland, project No. DEC-2014/15/B/ST4/04957", Charging/discharching mechanism at the electrode/electrolyte interface of supercapacitors decision of 11.05.2015.

**Conflicts of Interest:** The authors declare no conflict of interest.

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
