# Peer review of "Methods of Assessing Degradation of Supercapacitors by Using Various Measurement Techniques"

_applsci, doi:10.3390/app9112311_

Round 1

Reviewer 1 Report

The authors presented an investigation on the construction of supercapacitor samples based on thermographic measurements and noise emission.

Although I believe the paper is too qualitative, it is worthy for publication. My only major comment is about the method introduced on line 85. The authors are encouraged to describe that method adopted and remind secondary details to the reference. In addition they should give evidence about how the present work differs form that reference.

Author Response

Dear Reviewer,

We would like to thank you very much for comments regarding the article. Below please kindly find our reply.

Comment:

Although I believe the paper is too qualitative, it is worthy for publication. My only major comment is about the method introduced on line 85. The authors are encouraged to describe that method adopted and remind secondary details to the reference. In addition they should give evidence about how the present work differs form that reference.

Reply:

Kindly be advised that this article is one of the series describing the stages of the research on new constructions of supercapacitors. As a part of this paper both new supercapacitors as well as methods enabling their characterization were developed by applying various methods based on C and RESR measurements and also those based on noise measurements and temperature fields. The articles employing the results have been divided into separate announcements (articles) and therefore we have tried to avoid duplication of the presented set-ups and results in our publications.

According to the suggestion, the information on the main assumptions of the measurement system and algorithms used was included. In the measurement method, applied and described in the paper, no changes were introduced both in the scope of the measurement methodology and the algorithms used to determine the measured values. Since, during the conducted research, they were not changed, and this is the reason why the reference to the previous article was used.

Reviewer 2 Report

1, Literature review must be improved, it is important to describe if related work has been work and what has been done exactly, and what this work will convey to the readers.

2, As for thermography part, more theoretical description and analysis should be added. The authors should read and refer to previously conducted work in the field of thermography especially theories in order to access to the theoretical analysis part, such as Maldague group etc. In fact, I see some work related to thermography can also be found in the same journal. Please improve this part.

3, As for conclusion, following the previous suggestion, more analytical statements with theoretical explanation should be added, including imagery and graphical analysis. Now it seems more like an experimental report. It should convey more scientific sound to the authors.

Author Response

Dear Reviewer,

We would like to thank you very much for comments regarding the article. Below please kindly find our reply.

Comment:

We have allowed ourselves to consolidate comments 1, 2, 3.

1.      Literature review must be improved, it is important to describe if related work has been work and what has been done exactly, and what this work will convey to the readers.

2.      As for thermography part, more theoretical description and analysis should be added. The authors should read and refer to previously conducted work in the field of thermography especially theories in order to access to the theoretical analysis part, such as Maldague group etc. In fact, I see some work related to thermography can also be found in the same journal. Please improve this part.

3.      As for conclusion, following the previous suggestion, more analytical statements with theoretical explanation should be added, including imagery and graphical analysis. Now it seems more like an experimental report. It should convey more scientific sound to the authors.

Reply:

Kindly be advised that this article is one of the series describing the stages of the research on new constructions of supercapacitors. As a part of this paper both new supercapacitors as well as methods enabling their characterization were developed by applying various methods based on C and RESR measurements and also those based on noise measurements and temperature fields. The articles employing the results have been divided into separate announcements (articles) and therefore we have tried to avoid duplication of the presented set-ups and results in our publications.

In the article, we have tried to indicate that one of the suggested methods, in our opinion, is the one that allows to predict degradation of the tested supercapacitors. Our aim here was not to accurately show the theory, as we tried to describe it in the related articles that we refer to. We are currently preparing further publications that will be an attempt to interpret our results. The basic goal that we set ourselves in this publication was to illustrate that there is the possibility of predicting the occurrence of degradation based on noise measurements, which is the reason why the article may actually resemble a test report.

At the same time, we would like to thank you very much for indicating the publication of the Maldague group, especially because it is a part of the direction in which we develop our research (particularly in the reference to machine learning).

Round 2

Reviewer 1 Report

The authors addressed the only concern I expressed. I recommend the publication of the article in the present form.

Reviewer 2 Report

I do not have more comments, thanks.